# A Comparative DFT Study on Process Control Agents in the Mechanochemical Synthesis of PbTe

**DOI:** 10.3390/ijms231911194

**Published:** 2022-09-23

**Authors:** Hugo Rojas-Chávez, Alan Miralrio, José M. Juárez-García, Guillermo Carbajal-Franco, Heriberto Cruz-Martínez, Fernando Montejo-Alvaro, Manuel A. Valdés-Madrigal

**Affiliations:** 1Tecnológico Nacional de México, Instituto Tecnológico de Tláhuac II, Camino Real 625, Col. Jardines del Llano, San Juan Ixtayopan, Alcaldía Tláhuac, Mexico City 13550, Mexico; 2Tecnologico de Monterrey, Escuela de Ingeniería y Ciencias, Ave. Eugenio Garza Sada 2501, Monterrey 64849, Mexico; 3Industrial Division, Universidad Tecnológica del Estado de Querétaro, Av. Pie de la Cuesta 2501, Nacional, Santiago de Querétaro 76148, Mexico; 4Tecnológico Nacional de México, Instituto Tecnológico de Toluca, Division of Graduate Studies and Research, Av. Tecnológico s.n., Metepec 52149, Mexico; 5Tecnológico Nacional de México, Instituto Tecnológico del Valle de Etla, Abasolo S/N, Barrio del Agua Buena, Santiago Suchilquitongo 68230, Mexico; 6Tecnológico Nacional de México, Instituto Tecnológico Superior de Ciudad Hidalgo, Av. Ing. Carlos Rojas Gutiérrez 2120, Fracc. Valle de La Herradura, Ciudad Hidalgo 61100, Mexico

**Keywords:** semiconductor, DFT, process control agent

## Abstract

A process control agent is an organic additive used to regulate the balance between fracturing and mechanical kneading, which control the size of the as-milled particles. Tributyl phosphate (TBP) is evaluated to act as surface modifier of PbTe, and it is compared with the results obtained using formaldehyde (CH_2_O). In order to elucidate the nature of the interaction between TBP and the PbTe surface, global and local descriptors were calculated via the density functional theory. First, TBP and CH_2_O molecules are structurally optimized. Then, vertical ionization energies as well as vertical electron affinities are calculated to elucidate how both molecules behave energetically against removal and electron gain, respectively. The results were compared with those obtained from the electrostatic potential mapped on the van der Waals isosurface. It is inferred that the theoretical insights are useful to propose adsorption modes of TBP and CH_2_O on the PbTe surface, which are usable to rationalize the facets exposed by PbTe after the surface treatment. The optimized structures of the compound systems showed a close correlation between the surface energy shift (Δγ) and the PbTe facets exhibited. Finally, a Wulff construction was built to compare the usage of TBP and CH_2_O molecules in PbTe morphology.

## 1. Introduction

The high-energy milling (HEM) process induces high-energy impacts on the loaded powders in the vial by collisions between milling media and precursors, causing severe plastic deformation, repeated fracturing and mechanical kneading of particles [1,2,3], which have effects on the final structure, morphology and properties of the as-milled products [1,4]. For instance, it has been documented that in ductile–ductile systems, the mechanical kneading of particles is dominant during milling over fracturing [3,5]. After that, the as-milled particles tend to agglomerate, and, in some cases, they severely adhere to the milling media and vial [6]. For that reason, the balance between mechanical kneading and fracturing cannot be achieved, and consequently the mechanochemical synthesis is suppressed. Therefore, the Ostwald ripening effect (ORE) takes place leading to an increase in particle size and, in certain cases, considerable sticking of the powders to the milling tools [7,8].

The deagglomeration and stabilization of nanoparticles (NPs) during the HEM process can be achieved using different kinds of stabilizing agents, e.g., charged species, polymers or surfactants, among others. In this context, a process control agent (PCA) is an organic additive used to control the balance between fracturing and mechanical kneading of particles during the HEM process. PCAs are mostly organic molecules, which adsorb on the particle surface and thereby impede the clean particle-to-particle contact. Furthermore, it is documented that PCAs affect the energy transfer between the milling media and the precursors being milled [9], which controls the surface energy on the particles [10]. Even more, organic PCAs have been found to break down with the constituent elements during the HEM process, reacting with the raw powders [11]. Therefore, the usage of PCAs during milling has emerged as an efficient strategy not only to control the size and shape in NPs, but also to unravel structure–property relationships [1]. For that reason, approaches are offered to begin developing respective selection criteria and determine the correlations between different PCAs and milling outcomes, as has been reported in the literature [9].

To some extent, both physisorption and chemisorption processes seem to be clear in the stabilization of NPs during milling through organic molecules [12,13]. Because PCAs are the essential factor in maintaining stability and controlling facets in as-milled NPs, their effect on the as-milled NPs was analyzed by using experimental techniques [13,14], neural network models [15] and density functional theory (DFT) computations [1].

From an experimental point of view, the interactions at the organic–inorganic interface of the activated surface of the particle during the HEM process, are still a matter of discussion. Undoubtedly, there is a lack of theoretical and experimental information concerning which PCA properties are particularly important. Consequently, based on experimental findings, it is unclear how to select a proper PCA depending on the particular goals set for the preparation of the as-milled products. Therefore, in this work, an explanation based on global and local conceptual DFT descriptors is proposed, considering the chemical nature of the PCA molecule. Finally, a Wulff construction was built to compare the usage of tributyl phosphate (TBP) and formaldehyde (CH_2_O) molecules in PbTe morphology.

## 2. Results and Discussion

### 2.1. Surface Modifier Molecules

In order to compare how TBP and CH_2_O molecules behave as PCAs for lead telluride, both molecules were studied isolated and adsorbed on low-index PbTe surfaces. The structure of TBP was optimized by 20 generations of genetic algorithm process, within the molecular mechanics level of theory through the MMFF94 force field. Correspondingly, both TBP and CH_2_O were studied within the PBE-D2/USP level of theory. The optimized structures and electrostatic potential (ESP) are shown in Figure 1. It is important to keep in mind that ESP is produced by the total charge distribution around a molecule.

It is worth noting that is possible to compare the properties of TBP and CH_2_O within the frontier molecular orbital approach. The highest occupied molecular orbital (HOMO) of formaldehyde exhibits σ bonding contributions in the case of C–H bonds, being larger in the H atom, as well as a *p* orbital in the oxygen atom, see Figure 2. The above can be related to the preferential interaction between the H and O atoms of formaldehyde and lead telluride surfaces, which is congruent with data reported in the literature for this and other systems [1,16]. On the other hand, the lowest unoccupied molecular orbital (LUMO) of this molecule exhibits an antibonding π* orbital in the case of C–O bonds. In contrast, the noncarbon-bonded oxygen atom in the phosphate group of TBP shows a large *p* nonbonding contribution. Bonding σ orbitals were obtained in the case of the remaining oxygen atoms and carbon ones. Additionally, no contributions in phosphorus were observed in the LUMO of TBP, whereas massive ones were obtained, from the first members of tributyl chains to the unsaturated oxygen atom. In addition, *p* nonbonding contributions are exhibited in the case of several carbon atoms.

Following the previous analogy in terms of frontier molecular orbitals, the most important interactions in the case of TBP are expected to be obtained with the oxygen atom and the PbTe surface as well as through carbon atoms. Moreover, that region is the one with the lowest ESP and the most negative Bader’s charge of about −0.501 e. It is possible to assume that an important contribution to the ligand–surface adsorption energy arises from the initial electrostatic interaction between the phosphate group and the lead surface atoms. The above is due to the feasibility to carry out a nucleophilic attack, as previously stated, based on condensed Fukui indices. Since the TBP is considerably bigger, in comparison with the CH_2_O molecule, the dispersion interaction with the PbTe surface is expected to be higher as well.

Another important difference between TBP and CH_2_O molecules is their energetic properties. The energy gap of TBP is considerably larger than that computed for CH_2_O, denoting a more kinetically stable molecule, see Table 1. In contrast, the larger molecule obtained the lowest vertical ionization energy (VIE), making the donation of charge from TBP to the PbTe surface easier. The vertical electron affinity (VEA) of CH_2_O is almost twice that obtained for TBP, favoring its acceptance of electrons coming from the surface. Thus, it is feasible to suppose that the TBP molecule could interact more easily with the PbTe surfaces and even obtain larger binding energies.

### 2.2. Slab Model Analysis for TBP as Surface Modifier

In order to confirm the previous predictions, the TBP molecule interacting with the 3-layer slab models of low-index (100), (110) and (111)_Pb_ surfaces of lead telluride were fully optimized. The atoms at the bottom layer were fixed in their crystal positions. Subsequently, the TBP molecule fully optimized was set in eight different orientations, 1 Å above the surface. Analogous approaches were implemented recently to elucidate the behavior of sodium stearate interacting with calcium sulfate whiskers [17] as well as in dodecylamine and oleic acid on kaolinite surfaces [18].

The lowest energy structures were studied in depth; thus, up to three different structures were obtained and arranged in terms of relative energy, see Table 2. In the case of (100) surface, the TBP molecule achieved the highest ligand–surface adsorption energy of about −45.29 kcal·mol^−1^. Similarly, the TBP molecule on the (110) surface obtained up to −44.36 kcal·mol^−1^ of adsorption energy. However, the lowest E_ads_ was obtained in the case of the (111)_Pb_ surface of about −0.84 kcal·mol^−1^.

As shown in Figure 3, the (100) slab model exhibits the TBP molecule attached to the surface by the trybutil chain, with the shortest bond length Pb–C of about 3.462 Å, while other chains appear detached from the surface. Similarly, the (110) surface exhibits the TBP molecule adsorbed by the trybutil chain and with the shortest contact point of about 3.983 Å. In both cases, the chain is oriented to the carbon atoms, which is consistent with the high contributions coming from the *p* orbital. In contrast, the (111)_Pb_ surface interacting with the TBP molecule leads to a structure with a shorter bond length; in this case, the bond is Pb–O, which is computed as 2.980 Å.

In this case, the organic molecule is oriented by its remaining oxygen atom. Then, both predictions about the regions of preferential interaction were confirmed by structural optimization, see Figure 3.

From the energetic point of view, the most favorable surfaces to attach the TBP are (100) and (110), according to their adsorption energies. In addition, the calculation of the surface energy for the clean surfaces agrees with the following order: γ_100_ < γ_110_ < γ_110_. As shown in Table 3, the ratios taking into account the (100) face, according to the Wulff construction, are 1.5 and 1.8, respectively.

These ratios are fully altered by the introduction of TBP as the surface modifier. In the current case, the order is maintained but with different ratios; thus, values of 2.03 and 2.91 are obtained, see Table 4. Consequently, the most stable surface will be the (100) and consequently the most feasible to be exhibited experimentally in the case of PbTe particles.

The (110) surface must be detected with lower intensities than in the particles lacking surface modifiers. Even more, the (111)_Pb_ surface should be obtained with even lower intensity in the Bragg reflections. Consequently, a Wulff construction could be obtained to exemplify the possible shape of the nanoparticles produced by using TBP as a PCA, see Figure 4.

The contributions to the energy shift could be separated into two main aspects, namely the ligand–surface interaction energy, E^int^, and the surface distortion energy, E^dis^. Although the interaction energy is higher in the case of the (111)_Pb_ surface of about −32.29 kcal·mol^−1^, the distortion energy is higher for the (100)_Pb_ surface. The above is enough to stabilize more the (100) surface and to highlight its presence in the case of the PbTe nanoparticles. Finally, the differences among ligand–surface interaction energies can be explained by the charge transference, coming from the PbTe surface toward the TBP molecule. In terms of Bader’s charges, the highest charge transference to TBP was achieved in the case of the (100)_Pb_ surface, accounting for a charge of about −0.19 e, see Table 4.

## 3. Materials and Methods

Calculations were carried out within dispersion-corrected DFT at the PBE-D2/USP level through Quantum Espresso 6.1 package, as reported in the literature [1]. Dispersion-corrected DFT computations were performed to simulate the PbTe bulk structures and surfaces.

The following parameters were used through all calculations: Energy cut off of 50 Ry, kinetic energy cutoff for charge density and potential energy of 250 Ry, energy convergence threshold for self-consistency of 1 × 10^−6^ a.u., convergence criteria for geometry optimization on the total energy of 1 × 10^−4^ a.u. and 1 × 10^−3^ a.u. for ionic minimization. The structural optimization was performed by the quasi-Newton Broyden–Fletcher–Glodfarb–Shanno (BFGS) algorithm.

The unit cell of PbTe was originally set by means of atomic positions obtained from data reported in the literature [1,19]. On the other hand, formaldehyde and tributyl phosphate structures were optimized from those obtained by means of the Simplified Molecular Input Line Entry System (SMILES) retrieved from the Drugbank online database. In addition, tributyl phosphate was preoptimized at the molecular mechanics level of theory by using the MMFF94 force field by the genetic algorithm code for conformational search called the “Balloon”, version 1.8.2 [20]. Whereas the volume of the unit cell experimentally obtained for PbTe was ~383.953 Å^3^, it was reduced only by 0.71% in the case of the fully relaxed unit cell at the PBE-D2/USP level of theory. The unit cell was used for the convergence tests, finally setting the parameters described above. The Brillouin zone was scanned at the gamma point for structural optimizations.

In order to elucidate the nature of the interaction of TBP with the PbTe surface, global and local descriptors were calculated via the DFT. After the optimization of TBP and CH_2_O molecules, vertical ionization energies as well as vertical electron affinities were calculated to elucidate how both molecules behave energetically against removal and electron gain, respectively. Then, the results were compared with those obtained from the electrostatic potential mapped on the van der Waals isosurface.

## 4. Conclusions

The tributyl phosphate molecule was studied, isolated and interacted with the low-lying lead telluride surfaces. Differences with formaldehyde, as the PCA, were discussed as well. Furthermore, the full interaction between the tributyl phosphate molecule and all the surfaces was elucidated by means of global and local descriptors that determined the stability and reactivity of the PCA proposed. The dispersion-corrected DFT calculations show that the tributyl phosphate can modify the surface energy more significantly in comparison with formaldehyde. Additionally, a Wulff construction suggests the formation of faceted-like PbTe nanoparticles when TBP is used as a PCA.

## Figures and Tables

**Figure 1 ijms-23-11194-f001:**
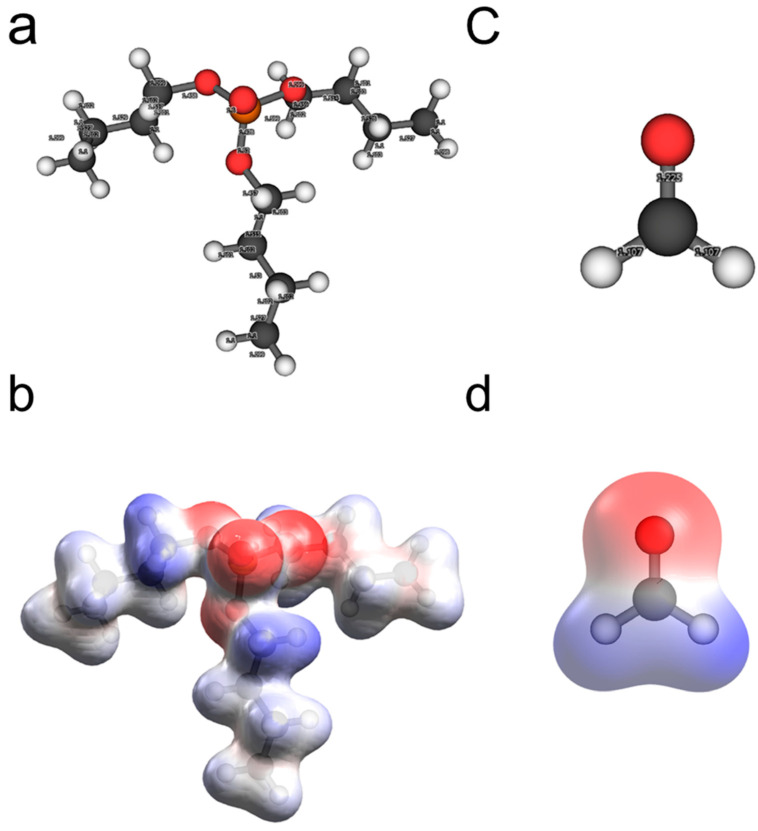
(**a**,**c**) Optimized structures for TBP and CH_2_O molecules. (**b**,**d**) ESP mapped on an isosurface with 0.004 a.u. of electron density for TBP and CH_2_O molecules. Note that color coding is used to represent the different values of the electrostatic potential; high ESP values are represented by blue color, and low ESP values are shown in red color, while white color represents the region of zero potential. White atoms are hydrogen; gray atoms are carbon; red ones are oxygen, and the yellow atom is phosphorus.

**Figure 2 ijms-23-11194-f002:**
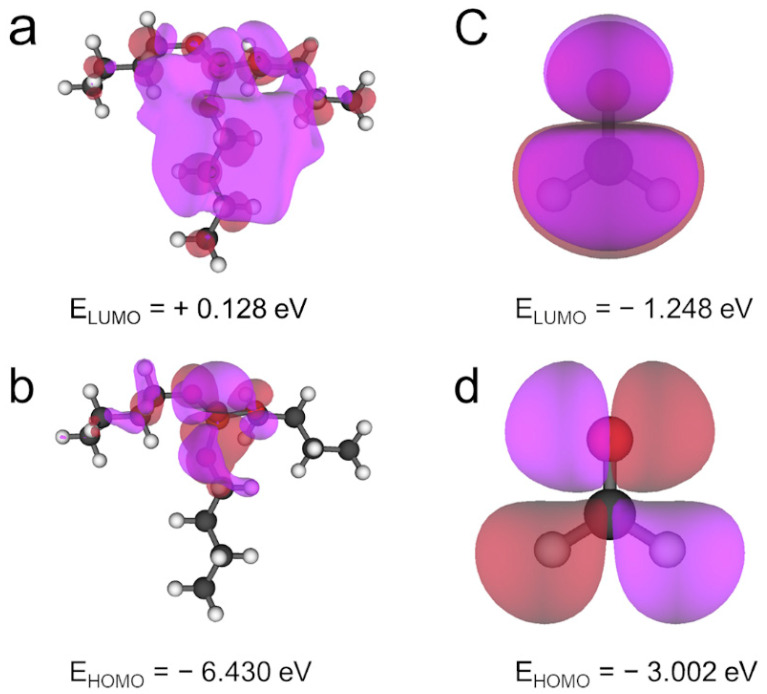
Frontier molecular orbitals for: (**a**,**b**) TBP molecule and (**c**,**d**) CH_2_O molecule. Isosurface with isovalue of 0.02 a.u.

**Figure 3 ijms-23-11194-f003:**
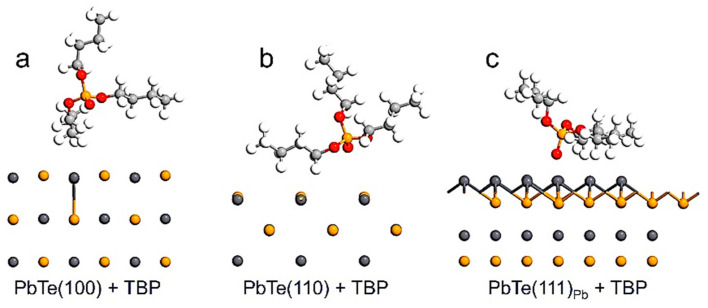
Ground state structures obtained for the TBP molecule interacting with the layer slab models of low-index (**a**) (100), (**b**) (110) and (**c**) (111)_Pb_ surfaces of lead telluride.

**Figure 4 ijms-23-11194-f004:**
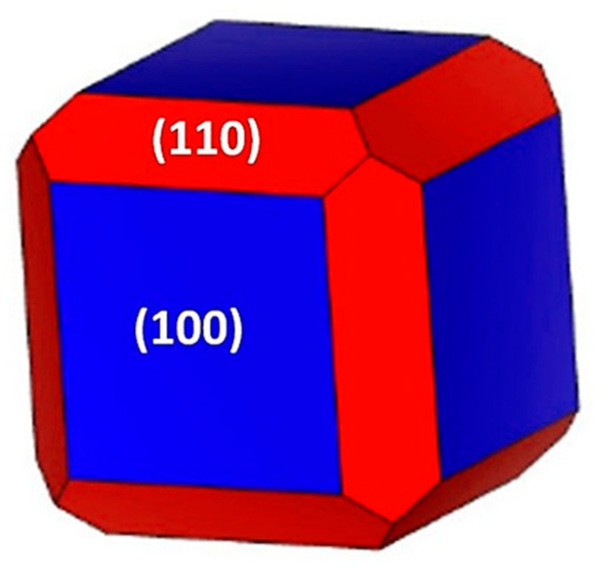
Wulff construction proposal for the shape of PbTe nanoparticles after using TBP as a PCA during milling.

**Table 1 ijms-23-11194-t001:** Vertical ionization energy (VIE), vertical electron affinity (VEA) and HOMO–LUMO energy gap for TBP and CH_2_O molecules.

Molecule	VIE (eV)	VEA (eV)	E_gap_ (eV)
TBP	8.272	–0.739	6.302
CH_2_O	9.936	–1.283	1.756

**Table 2 ijms-23-11194-t002:** Relative energy, ΔE, to the most stable surface. In addition, ligand–surface adsorption energy, E_ads_.

Surface	Structure	*Δ*E (kcal·mol^−1^)	E_ads_ (kcal·mol^−1^)
(100)	#1	0.00	−45.29
(110)	#1	0.00	−44.36
(110)	#2	1.75	−42.61
(110)	#3	0.96	−43.39
(111)_Pb_	#1	9.09	8.25
(111)_Pb_	#2	17.14	16.30
(111)_Pb_	#3	0.00	−0.84

**Table 3 ijms-23-11194-t003:** Total electronic energy, number of PbTe formula contained and area and surface energy γ obtained for (100), (110) and (111)_Pb_ clean surface models. Note that such computed values are in full agreement with [1].

Surface	n	Energy [Ry]	Area [Å^2^]	γ [meV·Å^−2^]
(100)	27	−4429.33	185.97	26.54
(110)	27	−4428.52	263.00	39.82
(111)_Pb_	18	−2952.41	139.36	46.91

**Table 4 ijms-23-11194-t004:** Interaction energy (E^int^), distortion energy (E^dis^), Bader’s charge in TBP, surface energy (γ), surface energy shift relative to unmodified surface (Δγ) and ratio for the low-lying states of (100), (110) and (111)_Pb_ surface models interacting with the TBP molecule.

Surface	Str	E^int^ [kcal·mol^−1^]	E^dis^ [kcal·mol^−1^]	Q_TBP_ [e]	γ [meV·Å^−2^]	Δγ [meV·Å^−2^]	Ratio to γ_100_
(100)	#1	−12.69	−32.60	−0.06	15.97	−10.57	1.00
(110)	#3	−13.11	−31.25	−0.07	32.50	−7.32	2.03
(111)_Pb_	#3	−32.29	−31.45	−0.19	46.64	−0.27	2.91

## Data Availability

Not applicable.

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
