# Peer review of "A Comparative DFT Study on Process Control Agents in the Mechanochemical Synthesis of PbTe"

_ijms, 2022, doi:10.3390/ijms231911194_

Round 1
Reviewer 1 Report
Dear Authors,
It was a pleasure to read the prepared article. First of all, the topic discussed was very interesting and extremely important in the context of HBM. The article is prepared with great precision and care. It will be my pleasure to follow further publications.
Regarding the merit of the article, I have no objections. However, in terms of editing, it seems to me it would be good to include uniform information for which substances (PCA) the calculations were performed. This is mentioned above in the Introduction section, but such information is mainly search in the Materials and Methods section.
It would be a good idea to review the article editorially because there are times when an unit flips to the next line or section titles are at the end of the page.
I wish you continued success in your studies and the best possible articles.
Best regards
Author Response
Comment: It was a pleasure to read the prepared article. First of all, the topic discussed was very interesting and extremely important in the context of HBM. The article is prepared with great precision and care. It will be my pleasure to follow further publications.
Regarding the merit of the article, I have no objections. However, in terms of editing, it seems to me it would be good to include uniform information for which substances (PCA) the calculations were performed. This is mentioned above in the Introduction section, but such information is mainly search in the Materials and Methods section.
It would be a good idea to review the article editorially because there are times when an unit flips to the next line or section titles are at the end of the page.
I wish you continued success in your studies and the best possible articles.
Response: We appreciate the minutious reading of our manuscript. With changes incorporated, we expect to reach the expectations of the International Journal of Molecular Sciences. Thank you very much.
Please note that the changes were highlighted in yellow color in the corrected version of the manuscript.
Reviewer 2 Report
Attached

Author Response
Attached you will find a point-by-point response to reviewer's comments.

Round 2
Reviewer 2 Report
The manuscript is fine now. It can be published in this form.